# The Accumulation of Electrical Energy Due to the Quantum-Dimensional Effects and Quantum Amplification of Sensor Sensitivity in a Nanoporous SiO_2_ Matrix Filled with Synthetic Fulvic Acid

**DOI:** 10.3390/s23084161

**Published:** 2023-04-21

**Authors:** Vitalii Maksymych, Dariusz Calus, Bohdan Seredyuk, Glib Baryshnikov, Rostislav Galagan, Valentina Litvin, Sławomir Bujnowski, Piotr Domanowski, Piotr Chabecki, Fedir Ivashchyshyn

**Affiliations:** 1Institute of Applied Mathematics and Fundamental Sciences, Lviv Polytechnic National University, Bandera Str. 12, 79013 Lviv, Ukraine; vitalii.m.maksymych@lpnu.ua; 2Faculty of Electrical Engineering, Czestochowa University of Technology, ul. J.H. Dąbrowskiego 69, 42-201 Częstochowa, Poland; dariusz.calus@pcz.pl (D.C.); piotr.chabecki@pcz.pl (P.C.); 3Faculty of Rocket troops and Artillery, Hetman Petro Sahaidachnyi National Army Academy, 32 Heroes of Maidan Street, 79026 Lviv, Ukraine; b.seredyuk@gmail.com; 4Department of Chemistry and Nanomaterials Science, Bohdan Khmelnytsky National University, Blvd. Shevchnko 81, 18031 Cherkasy, Ukraine; glibar@kth.se (G.B.); garol@ukr.net (R.G.); litvin_valentina@ukr.net (V.L.); 5Faculty of Mechanical Engineering, Bydgoszcz University of Science and Technology, 7 Kaliskiego Ave., 85-796 Bydgoszcz, Polandpiotr.domanowski@pbs.edu.pl (P.D.)

**Keywords:** encapsulate, silicon dioxide matrix, synthetic fulvic acid, impedance spectroscopy, photo- and magneto-resistive and capacitive effects, quantum battery

## Abstract

A heterostructured nanocomposite MCM-41<SFA> was formed using the encapsulation method, where a silicon dioxide matrix—MCM-41 was the host matrix and synthetic fulvic acid was the organic guest. Using the method of nitrogen sorption/desorption, a high degree of monoporosity in the studied matrix was established, with a maximum for the distribution of its pores with radii of 1.42 nm. According to the results of an X-ray structural analysis, both the matrix and the encapsulate were characterized by an amorphous structure, and the absence of a manifestation of the guest component could be caused by its nanodispersity. The electrical, conductive, and polarization properties of the encapsulate were studied with impedance spectroscopy. The nature of the changes in the frequency behavior of the impedance, dielectric permittivity, and tangent of the dielectric loss angle under normal conditions, in a constant magnetic field, and under illumination, was established. The obtained results indicated the manifestation of photo- and magneto-resistive and capacitive effects. In the studied encapsulate, the combination of a high value of ε and a value of the tgδ of less than 1 in the low-frequency range was achieved, which is a prerequisite for the realization of a quantum electric energy storage device. A confirmation of the possibility of accumulating an electric charge was obtained by measuring the I-V characteristic, which took on a hysteresis behavior.

## 1. Introduction

Nanotechnologies are being developed more and more intensively, not only in terms of the synthesis of and research into new nanomaterials, but new possibilities for their practical applications are also being studied to a greater extent [1,2]. Nanostructured materials, due to the nano-size of their structural components and the quantum nature of their phenomena, demonstrate not only much better characteristics (for example, increased strength, chemical reactivity, or conductivity), but also exhibit functional hybridity [3].

A careful study of such nanostructured materials has shown that a reduction in their physical dimensions from the microscopic to the meso- and nanoscopic scales leads to a change in most of their physical properties, such as their temperature and type of phase transitions [4,5,6,7,8,9], and a huge increase in their dielectric constant [10,11,12], etc. Additionally, these kinds of materials cause the quantum amplification of their sensory sensitivity to external physical fields [13,14,15].

Achieving a high dielectric constant due to the nanostructuring of the material will open up the possibility of accumulating electrical energy at the interphase heterojunctions. This essentially opens up the possibility of creating quantum analogs of autonomous power sources that will provide not only a significant increase in the energy-intensive characteristics of the latter, but will also allow for a significant increase in their speed and a direct incorporation into micro- and nanoelectronics. For example, researchers from the USA proposed using a thin dielectric layer with a large contact area, which was implemented in nanocomposites of ceramics and glass [16,17]. Specialists from Illinois University proposed a system of vacuum nanotubes and called it a ‘digital quantum battery’ [18]. A more modern work in this area is the theoretical model of a supercapacitor presented by scientists from Cornell University [19]. The researchers predicated two chains in their model: one chain dealt with electrons, whereas the other chain primarily dealt with holes. This study confirmed that their two-chain model demonstrated a significantly enhanced level of quantum capacitance. However, only the first steps have been taken in this direction, the results of which have more scientific significance than practical appeal.

Significant progress on this path can be achieved by forming inorganic/organic nanocomposites based on the host<guest> principle. The research carried out in this paper shows the promising nature of this type of materials for electrical energy storage devices at a quantum level [20]. The themes explored in this paper will be further developed in our subsequent scientific research.

## 2. Materials and Methods

Presently, there are many methods for the synthesis of nanostructured materials. One such method is the introduction of materials into porous matrices. This technological approach has a number of advantages:It makes it possible to obtain nanostructured materials with a wide range of controlled component sizes from ~1 nm to ~300 nm;It allows for the formation of nanostructured materials with different particle geometries and topologies: three-dimensional (3D) dendrites and regular structures, 2D film and layered structures, and 1D nanowires or 0D small nanoparticles;It has a large variability in the various heteroingredients for building nanostructured materials: metals, ferroelectrics, dielectrics, insulators, semiconductors, superconductors, and magnetic materials, as well as various organic compounds;It allows for the production of nanostructured materials in large quantities (up to several cubic centimeters). This, in turn, opens up the possibility of using some experimental methods that require a large amount of the studied material (for example, neutron scattering and heat capacity measurements, etc.).

As an inorganic dielectric matrix, a mesoporous regular structure based on the SiO_2_ matrix—MSM-41 (Mobil Composition of Matter No. 41 of the Sigma Aldrich trademark) was used. It is a matrix with ordered hexagonal channels that do not intersect. It is obtained by template technology. Its wall thickness ranges from 0.6 to 0.8 nm [21,22]. Its calibrated pore diameter can be purposefully varied in the range of 1.5–10 nm.

The guest component was synthetic fulvic acid (SFA), which was obtained via the oxidation of pyrocatechol (Sigma Aldrich, St. Louis, MO, USA) with molecular oxygen in an alkaline environment (Ph ≈ 13). The peculiarity of this SFA is that it has a chemical composition that is close to the natural one, i.e., containing N and S atoms. It is obtained by introducing thiourea (Sigma Aldrich) into the reaction mixture, which is a source of nitrogen and sulfur in synthesized fulvic acid. The choice of pyrocatechin and thiourea as precursors was due to their ability to enter into a substitution reaction, the product of which simultaneously contains a phenolic nucleus and thiocarbamide residue.

To prepare the reaction mixture, a 2.2 g weight of pyrocatechin in a polyethylene bag was placed in a 1 l Kjeldahl flask, then a solution containing 3.2 g of NaOH, 0.76 g of thiourea, and distilled water was carefully added to the wall, with a total volume of 100 mL. The air was purged from the flask by slowly blowing away the excess pure electrolytic oxygen. After the reactor flask was filled with oxygen, it was sealed and shaken mechanically. The shaking led to the mixing of the reagents and started the reaction. The reaction was carried out until the oxygen consumption slowed down to less than 0.01 mol O_2_ per 1 mol pyrocatechin per hour.

After the completion of the oxidation reaction, the reaction mixture was passed through an ion exchange column filled with KU-2-8 cationite in H-form. The removal of the eluate started when a brown color appeared and was completed when the color changed to a light red. The volume of the obtained fraction was equal to the volume of the reaction mixture. The liquid was heated on an electric stove to remove the excess dissolved carbon dioxide. The resulting solution had a pH of 2.8.

The substance was a black powder that was soluble in water. The aqueous solution had an acidic reaction (pH ≈ 3). From the point of view of the molecular structure, it was a complex mixture of oligomers containing –OH and –COOH groups. The average molecular weight was 860 g/mol. The substance gave an EPR signal, which indicated the presence of free radical centers in its structure, while the SFA synthesis technology was outlined in Ukrainian patents [23].

The encapsulation of the SFA in the pores of the MSM-41 matrix was carried out by the method of a thermovacuum impregnation of the powdered material of the matrix with the obtained SFA solution. We have effectively used this technique for the formation of encapsulates in previous studies [24]. The thermovacuum impregnation process consisted of the following stages: (1) the powder MSM-41 was heated to a temperature of 140 °C (413 K) and kept under reduced pressure for 2 h, in order to completely degas the pores; and (2) the next step, while maintaining the reduced pressure, was lowering the temperature to 45–55 °C (318–328 K) and introducing the SFA solution into the volume where the powder was located. After that, the powder was washed with distilled water and dried to a constant mass. Thus, the encapsulate MSM-41<SFA> was formed as a result of this thermovacuum intercalation technology.

The pore size value of the MSM-41 matrix that was selected for the research was determined by the standard method of an isothermal adsorption/desorption of nitrogen at its boiling temperature (T = 77 K), using the Quantachrome Nova Touch LX2 (USA) automated analyzer. Before these measurements were taken, the material samples were pre-degassed in a vacuum at 473 K for 16 h. The adsorption parameters were calculated based on the BET model.

In order to establish the structure of the obtained MSM-41<SFA> nanocomposite, several X-ray studies were conducted. The X-ray diffraction spectra were measured in scanning mode ϑ−2ϑ in Cu-Kα (λ=1.5419 Å) and radiation monochromatized by a reflection from the (111) plane of a Ge single crystal in the mode of X-ray beam passage. The measurement was carried out using a DRONE-3 (USSR).

The impedance spectra were measured to study the conductive and polarization properties. For this purpose, the formed powdery nanocomposite MSM-41<SFA> was pressed into the form of a tablet with a diameter of 6.2 mm, a thickness of 0.96 mm, and a weight of 44 mg, using a mold. Ohmic contacts were deposited onto the parallel planes of the tablets using a silver contact paste.

Research utilizing the method of impedance spectroscopy was carried out in the frequency range of 10^−3^–10^6^ Hz, using the measuring complex “AUTOLAB” of the Dutch “ECO CHEMIE” company, equipped with the computer programs FRA-2 and GPES. The amplitude of the measuring signal was 0.010 V. The ambiguous points were removed using a Dirichlet filter [25]. The impedance parameters were measured under normal conditions, in a constant magnetic field with a strength of 220 kA·m^−1^ and, when illuminated by a solar radiation simulator for the standard AM 1.5 G solar spectrum, the total available power was 982 W/m^2^. The external fields were superimposed in the direction of the measurements of the impedance spectra. This measurement geometry was chosen in order to be able to omit the Lorentz force.

The current–voltage characteristics of the obtained structure were also measured in the range of −3 ÷ +3 V, with a potential change rate of 0.050 V·s^−1^. The measuring range and scanning speed were selected based on the experimental experience of studying these types of samples, in order to avoid damage.

The structure of the energy impurity spectra was also studied by the method of a thermally stimulated discharge in the temperature range of −25–70 °C, with a constant heating rate of 5 degrees/min. The thermally stimulated discharge spectra were recorded in the mode of short-circuited contacts. To measure the current, the B7-30 (USSR) was used, which is characterized by an input impedance of 10^15^ Ohm.

## 3. Results and Discussion

Using the experimental method of nitrogen sorption/desorption, using the Quantachrome Nova Touch LX2 automated analyzer, it was established that the studied matrix MSM-41 is characterized by a high degree of monoporosity, with a maximum in the distribution of its pores by a radius of—1.42 nm (Figure 1).

In the X-ray diffraction pattern that was measured for the MSM-41 matrix, we obtained the usual picture—a broad diffuse maximum, which is a characteristic of an amorphous material. In the case of the MSM-41<SFA> encapsulation, practically the same picture was obtained (Figure 2). The absence of peaks from the SFA can be explained by its nanodispersity, as a result of which, they were not visible against the background of the high intensity of the main diffuse maximum of the MSM-41 matrix.

The next step was to study the conductive and polarization properties of the synthesized encapsulant MSM-41<SFA>. As a result of the conducted research, the impedance spectra were experimentally measured for the original MSM-41 matrix and for the MSM-41<SFA> encapsulate.

That is, the frequency dependences of the total complex resistance $\hat{Z}$ can be represented by the ratio:(1)$$\hat{Z}\left(\omega \right)=Z^{\prime }\left(\omega \right)+{\mathrm{iZ}}^{\prime \prime }\left(\omega \right)$$
where $Z^{\prime }$ and $Z^{\prime \prime }$ are, respectively, the real and imaginary parts of the complex resistance.

In order to find out the conductive properties, we analyzed the frequency behavior of $Z^{\prime }(\omega )$ for the original matrix MSM-41 and the encapsulated MSM-41<SFA> (Figure 3).

The corresponding dependencies are shown in Figure 3 as curve 1 and 2. Analyzing their behavior makes it is clear that the original investigated matrix MSM-41 is characterized by a monotonically decreasing character of $Z^{\prime }\left(\omega \right)$ and the dependence of the resistance on the frequency within the entire investigated range. In this case, the conductivity of the MSM-41 matrix can be written according to the known ratio [26]:(2)$$\sigma^{\prime }=\sigma_{dc}+A\omega^{n},$$
here, $\sigma_{dc}$ is the specific conductivity measured at the direct current at a given temperature by the available band current carriers, and A and n are the temperature- and composition-dependent parameters. According to the traditional approach, the first term in (3) is determined as:(3)$$\sigma_{dc}=en\mu ,$$
where e is the electron charge, n is its concentration, and μ is its mobility. The second term in (3) represents the polarization component of the total conductivity, which is mainly formed by the hopping of the charge carriers in localized states near the Fermi level, or by the excitation processes of their capture in the tails of the bands or energy bands of non-localized states.

Thus, the first term in (2) reflects the conductivity formed by the band carriers, which does not depend on the frequency, and the second term in (2) reflects the conductivity formed by the hopping conductivity, which depends on the frequency. Accordingly, from the dependence of the $Z^{\prime }(\omega )$ on the MSM-41 matrix, we can see the suppressive influence of the sudden change in the conductivity over the band conductivity within the entire studied range.

The introduction of SFA into the pores of the MSM-41 matrix leads to an almost five-fold decrease in the $Z^{\prime }$ in the low-frequency range (0.004–70 Hz) (curve 2 in Figure 3), and subsequently, more than a four-fold increase at higher frequencies (70–5.5·10^4^ Hz). The nature of the behavior of the $Z^{\prime }(\omega )$ also changes, which, for the MSM-41<SFA> encapsulation, means that it loses its monotony in the medium-frequency range (0.8–70 Hz) and exhibits oscillatory behavior. In this case, relation (2) is no longer valid. The deviation in the experimental dependence is due to the fact that the conductivity of this encapsulate is formed not only by band carriers and jumps in localized states, but also by the process of the capture–retention–release of carriers from quantum wells.

In general, the behavior of the $Z^{\prime }\left(\omega \right)$ for the MSM-41<SFA> encapsulate indicates significant changes in the concentration of the impurity states near the Fermi level by the guest component. In this case, shallow trap centers are also formed, which capture and hold the carriers for a time that is proportional to the period of the sinusoidal measurement signal, thus leading to oscillations of the $Z^{\prime }\left(\omega \right)$ [24].

The imposition of a constant magnetic field leads to a significant decrease in the $Z^{\prime }$ and an increase in the amplitude of medium-frequency oscillations (curve 3 in Figure 3). The decrease in the $Z^{\prime }$ shows that the change in the resistance is not related to a decrease in the mobility of the current carriers in a constant magnetic field, but that it is caused solely by an increase in the concentration of free charge carriers due to the Zeeman effect, which results in the delocalization of a significant number of charge carriers. It becomes obvious that the guest subsystem is responsible for the increase in the concentration of the free carriers in the constant magnetic field. The constant magnetic field also activates additional quantum wells, which leads to an increase in the amplitude of medium-frequency oscillations. An interesting fact is that the magnetoresistive effect is observed not only at low frequencies, but also in the high-frequency region. The magnetoresistive effect is calculated by the formula $\chi_{H}=\rho_{0}/\rho_{H}$. $\rho_{0}$ is the value of the resistance Z′ under normal conditions, and $\rho_{H}$ is the value of the resistance Z′ when a constant magnetic field is applied. Its maximum values obtained at a frequency of 0.001 Hz take the value of $\chi_{H}$ ≈ 3 times, at a frequency of 0.8 Hz $\chi_{H}$ ≈ 4 times, at a frequency of 1 kHz $\chi_{H}$ ≈ 4 times, and at a frequency of 1 MHz $\chi_{H}$ ≈ 25 times. The obtained result has an important practical significance for the manufacture of ultrasensitive magnetic field sensors.

The effect of illumination also leads to a decrease in the $Z^{\prime }\left(\omega \right)$ (curve 4 in Figure 3), which, in this case, is the result of the photoexcitation of carriers from the impurity energy levels. At the same time, deeper trap centers are activated, leading to additional oscillations of the $Z^{\prime }\left(\omega \right)$ in the lowest frequency range. In this case, the photosensitivity is also most likely caused by a guest energy subsystem formed in a unique way. The photoeffect manifests itself under the condition of an increasing frequency. The photoeffect is calculated by the formula $\chi_{L}=\rho_{d}/\rho_{l}$. $\rho_{d}$ is the resistance value Z′ under normal conditions in the dark and $\rho_{l}$ is the resistance value Z′ under illumination. At a frequency of 0.8 Hz, the value $\chi_{L}$ ≈ 4 times, at a frequency of 1 kHz $\chi_{L}$ ≈ 4 times, and at a frequency of 1 MHz $\chi_{L}$ ≈ 9 times. The obtained result shows the prospects of using this MSM-41<SFA> encapsulation for the manufacture of highly sensitive light sensors and elements for the conversion of a light signal into an electrical signal.

In order to investigate in more detail the structure of the energy spectrum of the guest impurity subsystem, thermally stimulated discharge currents were measured (Figure 4). No thermally stimulated discharge currents were recorded for the original MSM-41. For MSM-41<SFA>, as we can see (Figure 4), they were recorded in the temperature range of 247–315 K. The spectrum has narrow strips with a significantly higher density of its states and a well-defined miniband character. The spectrum acquires quasi-continuity at temperatures higher than room temperature, starting from 315 K. The relaxation of the homocharge is observed throughout the entire studied temperature range.

The next step is to consider the behavior of the impedance hodograph, presented in the form of Nyquist diagrams that are shown in Figure 5. The Nyquist diagrams for both the original MSM-41 matrix and the MSM-41<SFA> encapsulate are two clearly defined semicircles that reflect the transfer of the electric charge through the MSM-41 matrix itself and between the matrix particles. When modeling, such an impedance dependence can be represented by the serial connection of two parallel R‖C links (inset to Figure 5).

The first semicircle from the origin of the coordinates (link R_1_‖C_1_), which corresponds to the high-frequency part of the spectrum, represents the processes of current flow through the volume of the material’s particles, and the second semicircle (R_2_‖C_2_), which corresponds to the low-frequency part of the spectrum, represents the processes of current flow in the space between the particles. After conducting simulations in the ZView2 software environment of the Nyquist diagrams of the corresponding equivalent electrical circuits, the following element values were obtained (Table 1). It is evident that SFA intercalation leads to a decrease in the resistances of R_1_ and R_2_. The capacity of C_1_ practically does not change. However, the capacitance C_2_, the value of which increases by an order of magnitude, changes greatly. This indicates an increase in the polarization of the MSM-41<SFA> encapsulated particles and the accumulation of an electric charge on their boundaries. The action of a permanent magnetic field and illumination leads to an increase in the conductivities and capacities of both the areas of current flow.

The obtained results indicate that the imaginary particles of the complex resistance will also undergo significant changes. This can be written by the following ratio:(4)$$Z^{\prime \prime }\left(\omega \right)=-\frac{1}{\omega C}$$

The formula for the capacity of a flat capacitor is as follows:(5)$$C=\frac{\varepsilon \varepsilon_{0}S}{d}$$

Therefore, on the basis of the experimentally determined values of the Z″, using Equations (4) and (5), it is possible to determine the value of ε.

The behavior of ε(ω), for a more detailed understanding of the polarization properties of the encapsulant MSM-41<SFA>, is analyzed in Figure 6. The anomalous behavior of the dielectric constants for all the measurements should be noted. From the point of view of the practical significance [17] of the obtained results, it is interesting to consider the high values of the ε in the frequency intervals, where the tangent of the dielectric loss angle is tgδ < 1 (Figure 7). This condition corresponds to the low-frequency part of the frequency spectrum. The introduction of the guest component leads to an increase in the ε(ω) in the frequency range of 2·10^−3^ ÷ 3·10^−1^ Hz (curves 1 and 2 in Figure 6), taking maximum values greater than 10^4^. Such behavior of the ε(ω) most likely arises due to Maxwell–Wagner segmental polarization and the additional polarization that occurs when the charge carriers jump over the localized states near the Fermi level. The Maxwell–Wagner polarization occurs in the presence of macrodipoles that arise in the vicinity of charged defects, as well as at the heteroboundaries of the structure. In this case, instead of one group of particles being capable of relaxation polarization with one defined relaxation time τ, there is a large number of relaxers with different values of the time τ. This is confirmed by the dependence of the tgδ(ω) (curves 1 and 2 in Figure 7). For the original matrix MSM-41 (curve 1 in Figure 7), a pronounced main peak is observed that corresponds to the main relaxation time for this material. On the other hand, in the case of the MSM-41<SFA> encapsulation, a tgδ(ω) oscillation is obtained, which clearly indicates the existence of a wide range of relaxation times caused by the relaxation centers introduced by the guest component.

This is confirmed when a constant magnetic field is applied, as a result of which, the value of the ε increases strongly (curve 3 in Figure 6) but the condition tgδ < 1 becomes no longer valid (curve 3 in Figure 7). On the other hand, illumination also leads to an increase in the ε (curve 4 in Figure 6), but the condition tgδ < 1 is fulfilled (of 2·10^−3^ ÷ 3·10^−1^ Hz) (curve 4 in Figure 7). In this case, the photocapacitance effect $\xi_{L}=\varepsilon_{d}/\varepsilon_{l}$ reaches a three-fold value (of 2·10^−3^ ÷ 3·10^−1^ Hz). $\varepsilon_{d}$ is the value of the dielectric constant under normal conditions in the dark and $\varepsilon_{l}$ is the value of the dielectric constant under illumination. In the high frequency range of 10^2^–10^6^ Hz, $\xi_{L}=\xi_{H}\approx$ five times.

The obtained result shows the opportunity of the MSM-41<SFA> encapsulate to be used as a material for the manufacture of quantum battery storing electric energy. Controlling its parameters with the help of light allows for the creation of its functional hybridity. The essence of the concept is to store electrical energy using polarized electrons and holes, rather than in the form of electrons and electrolyte ions, as in the case of an electrochemical supercapacitor. Known dielectrics which have this ability display a ε ≤ 100 property. The author of the patent [17] theoretically showed that, in the case of increasing the ε to ~10^6^, it is possible to achieve a higher energy density of 1 MJ/kg. At the moment, these are theoretical calculations that allow us to see the prospects for development in this area of research.

A confirmation of the ability of the encapsulate MSM-41<SFA> to accumulate an electric charge is obtained by measuring the I-V curve (Figure 8). In this case, for the MSM-41<SFA> encapsulate, the I-V curve takes a form different to a linear one, which is typical for the original desorbed matrix MSM-41, reflecting the hysteresis that is characteristic of non-Faraday energy storage devices (curve 1 in Figure 8). This is typical for such devices, for example, for supercapacitors, which work on the effect of a double electric layer, where a charge separation takes place at the solid–electrolyte boundary. In our case, the accumulation of an electric charge occurs at the grain boundaries due to polarization.

The observed noise on the I-V curve is most likely caused by the electron–phonon interaction, which obviously takes place while taking into account the complex quantum mechanisms of the current flow described above.

This result opens up the prospect of creating non-electrochemical sources of energy storage that will allow for an increase in their specific energy-intensive indicators and facilitate their incorporation directly into the structure of micro- and nanoelectronics.

Illumination leads to an order of magnitude increase in the I-V current (curve 3 in Figure 8), and a constant magnetic field leads to a 30-fold increase in the I-V current (curve 2 in Figure 8). At the same time, in both cases, the hysteresis loop narrows significantly. It should also be noted that, in a constant magnetic field, the noise of the I-V curve is preserved and is practically invisible under illumination.

In order to correctly compare our results with the data presented in [17], we calculated the specific capacitance for both cases. The paper indicates such parameters as the specific energy W = 1.66 MJ/kg when the voltage on the capacitor covers reaches 1.37·10^5^ V. The energy of a capacitor is determined by the formula:(6)$$W=\frac{CU^{2}}{2}$$
where C is the capacitance of the capacitor, and U is the voltage across the capacitor.

The capacitance will be determined accordingly:(7)$$C=\frac{2W}{U^{2}}$$

As a result of simple calculations, the specific capacitance C = 1.77·10^−4^ (F/kg) = 1.77·10^−7^ (F/g) = 0.177 (μF/g).

Accordingly, we can calculate the specific capacity of our material by using the formula:(8)$$C=\frac{q}{Um}$$
where q is the accumulated charge on the capacitor linings, U is the voltage across the capacitor, and m is the mass of the sample under testing.

From the presented I-V current in Figure 8, we can determine the value of q by integrating the current over time
(9)q=∫Idt
for the measurements under normal conditions in a constant magnetic field and under illumination. Substituting these values into formula (8), we obtain the following values for the specific capacitance of our material: C_nc_ = 0.053 μF/g, C_m_ = 1.260 μF/g, C_l_ = 0.56 μF/g. Therefore, from the data presented, we can see that our material has more than twice the specific capacitance that was theoretically proposed in [17], but that it has a functional hybridity and takes on a higher value when illuminated, and a much higher value when a constant magnetic field is applied.

## 4. Conclusions

An MSM-41<SFA> encapsulation was formed for the first time. The introduction of SFA into the pores of the MSM-41 matrix led to an almost five-fold decrease in the $Z^{\prime }$ in the low-frequency range (0.004–70 Hz) (curve 2 in Figure 3), and subsequently, more than four-fold increase at higher frequencies (70–5.5 10^4^ Hz). The conductivity of this encapsulate was formed not only by band carriers and jumps in localized states, but also by the process of the capture–retention–release of carriers from quantum wells.

The impurity energy spectrum of the MSM-41<SFA> encapsulate, in the temperature range of 247–315 K, represented narrow bands with a significantly higher density of states and a well-defined miniband character, where above 315 K, it became quasi-continuous. At the same time, a homocharge relaxation was observed.

For the MSM-41<SFA> encapsulation, a magnetoresistive effect was observed, the maximum values of which reached: at a frequency of 0.001 Hz $\chi_{H}$ ≈ 3, at a frequency of 0.8 Hz $\chi_{H}$ ≈ 4 times, at a frequency of 1 kHz $\chi_{H}$ ≈ 4 times, and at a frequency of 1 MHz $\chi_{H}$ ≈ 25 times. A photoresistive effect was also observed: at a frequency of 0.8 Hz, it took the value $\chi_{L}$ ≈ four times, at a frequency of 1 kHz, $\chi_{L}$ ≈ four times, and at a frequency of 1 MHz, $\chi_{L}$ ≈ nine times. The photoresistive effect reached the value of $\xi_{L}\approx$ three times. These obtained results have important practical significance for the manufacture of ultrasensitive magnetic field sensors and resistive- and capacitive-type lighting.

In the MSM-41<SFA> encapsulation, it was shown that, in the frequency interval of 2·10^−3^–3·10^−1^ Hz, the following values were reached: tgδ < 1, ε ≈ 10^4^, which opens up the possibility of implementing the quantum storage of electrical energy. At the same time, the value of the ε noticeably increased under the influence of illumination, ensuring the functional hybridity of such a quantum battery.

The I-V curve of the MSM-41<SFA> encapsulation assumed a hysteresis characteristic of devices with the capacitive storage of an electric charge. In our study, we showed that there is a separation between electric charges and their accumulation at interphase boundaries.

## Figures and Tables

**Figure 1 sensors-23-04161-f001:**
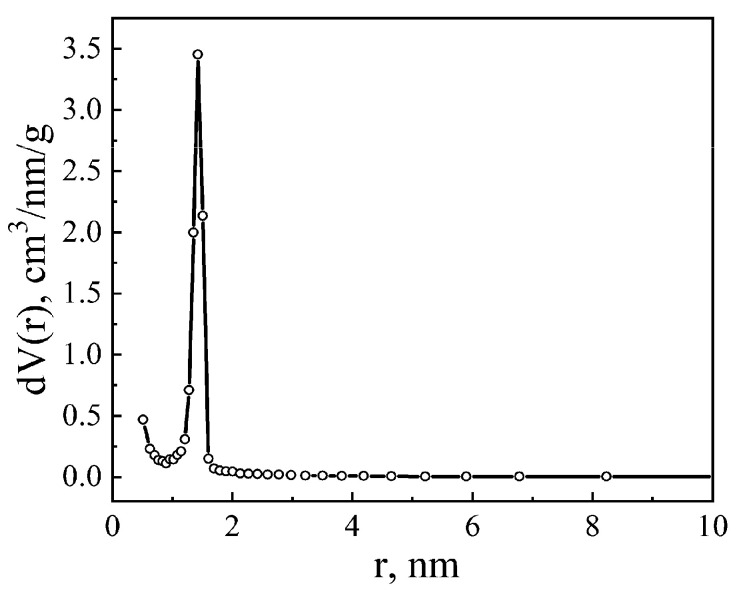
Distribution of pores in MSM-41 according to the adsorption branch of the isotherm, according to the BET method.

**Figure 2 sensors-23-04161-f002:**
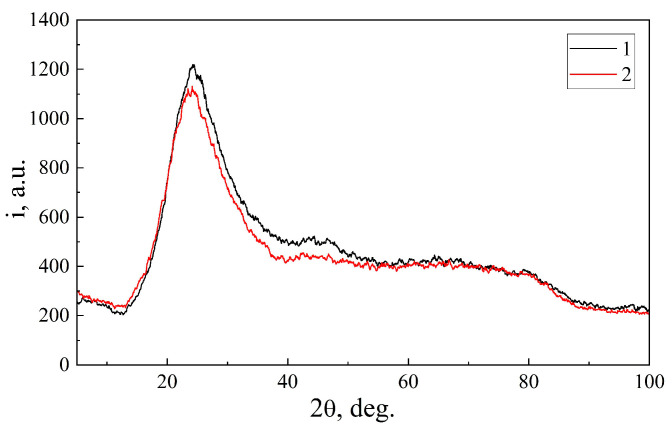
XRD diffraction patterns of: (1) MSM-41 matrices, and (2) MSM-41<SFA encapsulate.

**Figure 3 sensors-23-04161-f003:**
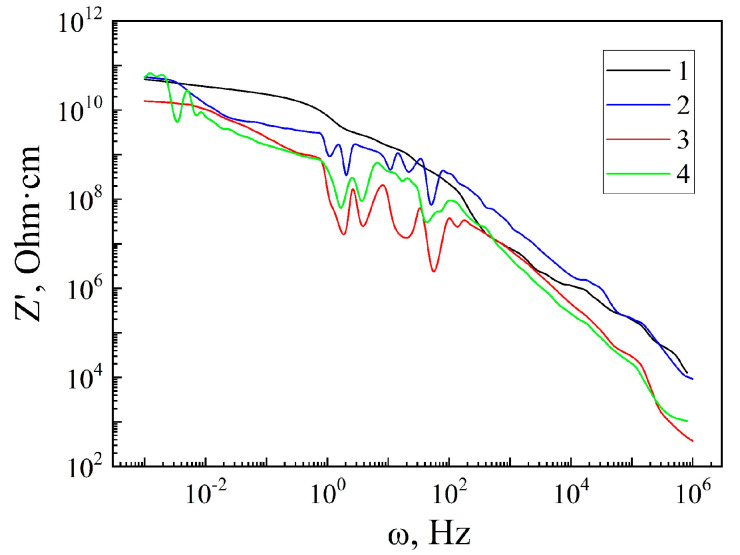
Frequency dependences of the real component of the specific impedance of the matrix MSM-41 (1) and encapsulate MSM-41<SFA>, measured under normal conditions (2), in a magnetic field (3), and under illumination (4).

**Figure 4 sensors-23-04161-f004:**
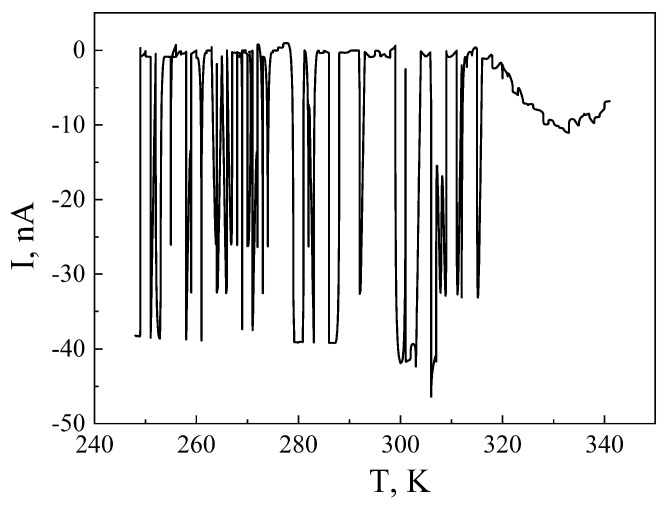
Thermally stimulated discharge currents for MSM-41<SFA> encapsulation.

**Figure 5 sensors-23-04161-f005:**
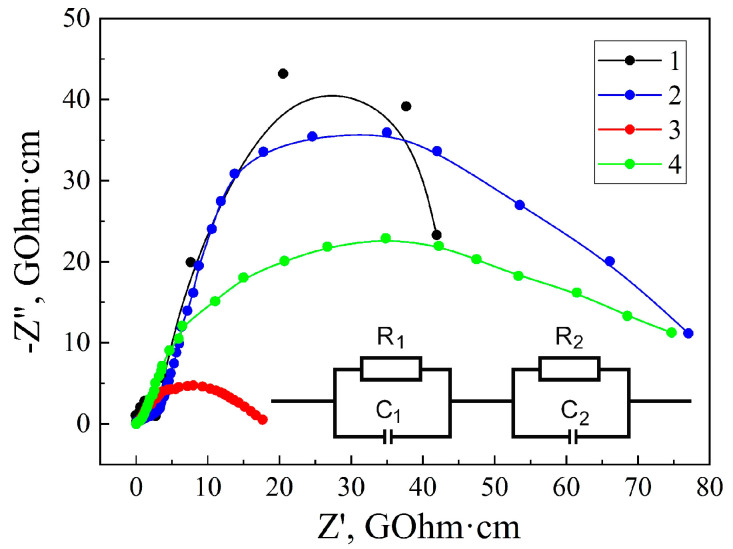
Nyquist diagrams measured for the matrix MSM-41 (1) and encapsulate MSM-41<SFA> under normal conditions (2), in a magnetic field (3), and under illumination (4).

**Figure 6 sensors-23-04161-f006:**
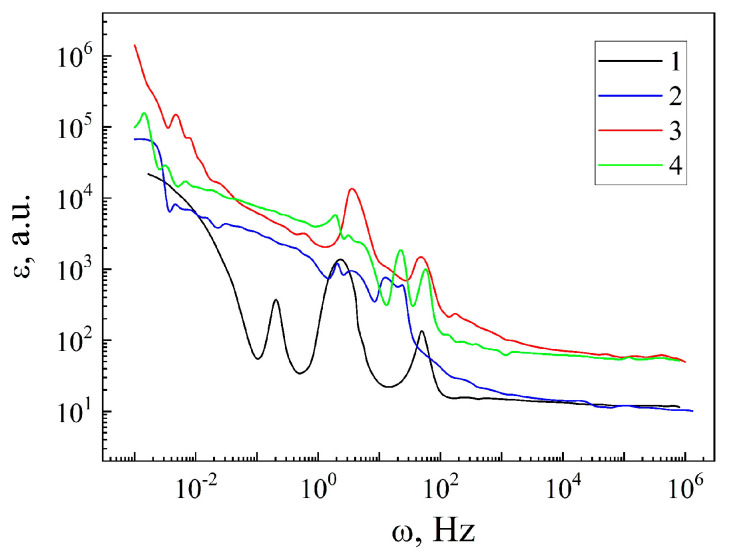
Frequency dependences of the dielectric constants of the matrix MSM-41 (1) and encapsulate MSM-41<SFA>, measured under normal conditions (2), in a magnetic field (3), and under illumination (4).

**Figure 7 sensors-23-04161-f007:**
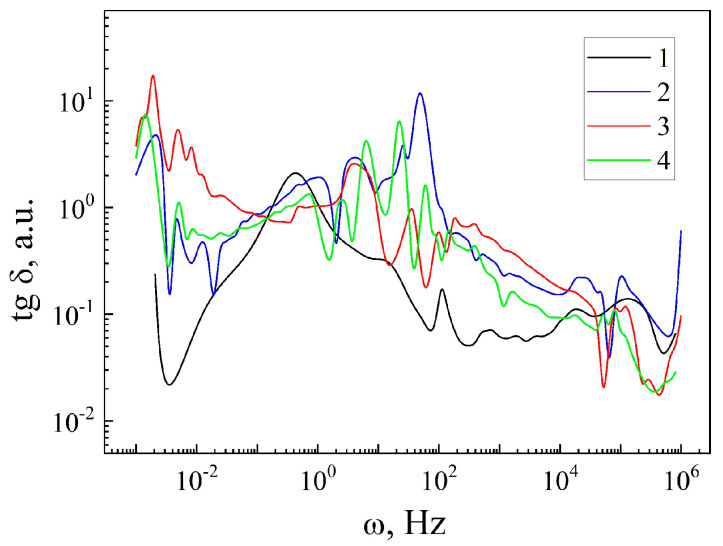
Frequency dependences of the tangent angles of dielectric losses of the matrix MSM-41 (1) and encapsulate MSM-41<SFA>, measured under normal conditions (2), in a magnetic field (3), and under illumination (4).

**Figure 8 sensors-23-04161-f008:**
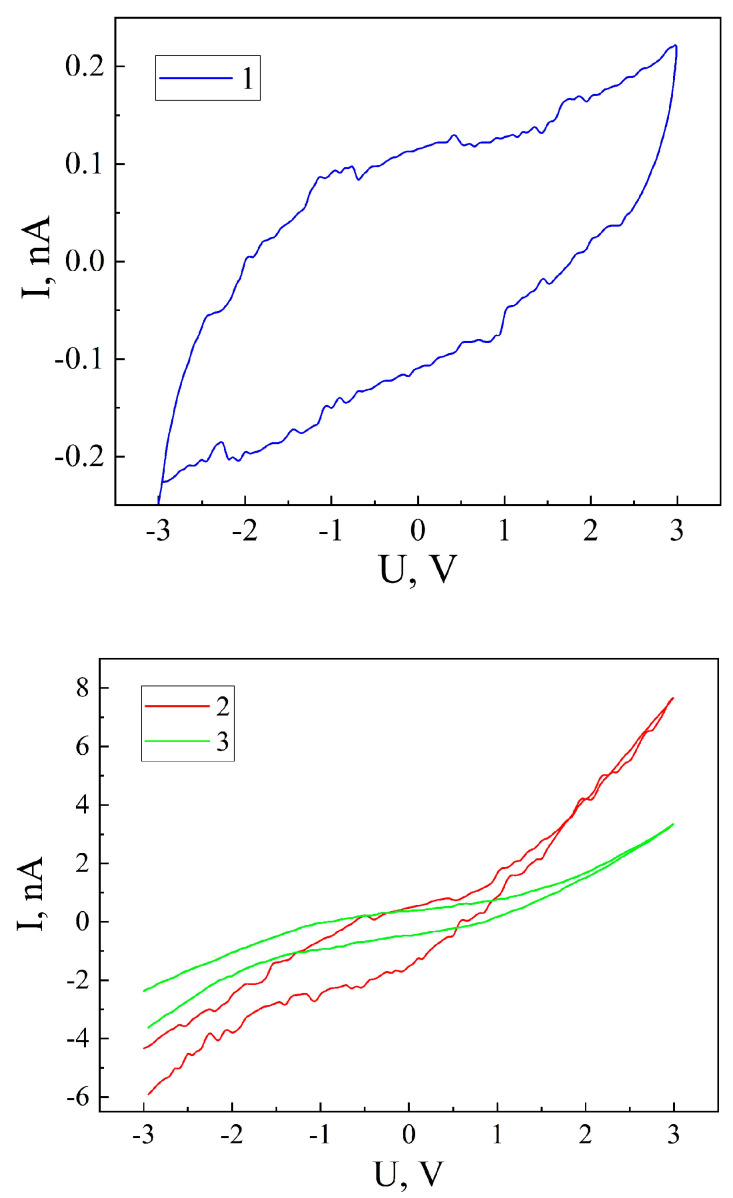
I-V curves of encapsulate MSM-41<SFA>, measured under normal conditions (1), in a magnetic field (2), and under illumination (3).

**Table 1 sensors-23-04161-t001:** Parameters of an equivalent electric circuit for MCM-41 matrix and MCM-41<SFA> incapsulate.

	R_1_, Ohm	C_1_, F	R_2_, Ohm	C_2_, F
MCM-41	2.978 × 10^9^	1.379 × 10^−12^	6.650 × 10^10^	1.024 × 10^−11^
MCM-41<SFA>_nc_	2.057 × 10^9^	1.348 × 10^−12^	6.002 × 10^10^	1.0346 × 10^−10^
MCM-41<SFA>_mf_	4.644 × 10^8^	7.981 × 10^−12^	1.001 × 10^10^	4.090 × 10^−10^
MCM-41<SFA>_l_	6.822 × 10^8^	6.464 × 10^−12^	5.485 × 10^10^	7.233 × 10^−10^

## Data Availability

Not applicable.

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
