# Peer review of "The Accumulation of Electrical Energy Due to the Quantum-Dimensional Effects and Quantum Amplification of Sensor Sensitivity in a Nanoporous SiO2 Matrix Filled with Synthetic Fulvic Acid"

_sensors, 2023, doi:10.3390/s23084161_

Round 1
Reviewer 1 Report
This paper work appear sound work and novelty however for wider readership this paper needs modification
1. Abstract must be freeze and deal only work done
2. Introduction must be correlated with new recent references alongwith mapped with recent work
3. Figure quality must be improved
4. In I-V characteristic looks noisy ? A better explanation and recorded curve must incorporate
5. In result and discussion text must be in running not pointwise
6. References part may need polishing
7. For comparative study Table must contain other group reported data
Author Response
- Abstract must be freeze and deal only work done.
Annotation has been improved
- Introduction must be correlated with new recent references alongwith mapped with recent work.
The introduction has been improved. Replaced references 10, 11 and 12 with newer ones, and added references 13 and 14 on sensing. Removed self-citations, leaving only reference 17, which was also replaced by a newer one.
- Figure quality must be improved.
Figures 2 and 4 were indeed of low resolution. They have been improved.
- In I-V characteristic looks noisy ? A better explanation and recorded curve must incorporate.
Added to the methodology «The current-voltage characteristics of the obtained structure were also measured in the range of -3÷+3 V with a potential change rate of 0.050 V·s⁻¹. The measuring range and scanning speed were selected based on the experimental experience of studying this type of samples in order to avoid damage.»;
and in the text of the explanation to Figure 8 «The observed noise on the I-V curve is most likely caused by the electron-phonon interaction, which obviously takes place taking into account the complex quantum mechanisms of current flow described above.»
- In result and discussion text must be in running not pointwise.
Сhanges in the text for better presentation of the materialю
- References part may need polishing.
Links have been carefully reviewed
- For comparative study Table must contain other group reported data.
The tables in this case are informational, not comparative.
Thank you for the review,
і respectfully Fedir Ivashchyshyn
Reviewer 2 Report
The article entitled “Accumulation of Electrical Energy due to Quantum-Dimensional Effects and Quantum-Amplification of Sensor Sensitivity in a Nanoporous SiO2 Matrix Filled with Synthetic Fulvic Acid” can be accepted for the publication after minor revision. Please find my comments below.
1. What is the method used to form the heterostructured nanocomposite MCM-41, and what are the respective roles of the host matrix and the organic guest in the composite material?
2. How is the monoporosity of the studied matrix established, and what is the maximum radius of the distribution of pores observed in the matrix?
3. What are the results of the X-ray structural analysis of the matrix and encapsulate, and what does the absence of manifestation of the guest component suggest about its nature?
4. What are the findings of the impedance studies of the encapsulate, and what does the frequency behavior of its impedance, dielectric permittivity, and tangent of the dielectric loss angle indicate about the encapsulate's properties?
5. What are the potential applications of the photo- and magneto-resistive and capacitive effects observed in the encapsulate, and how can they be utilized in the preparation of highly sensitive sensors?
6. How are the conditions for quantum storage of electrical energy implemented in the nanocomposite, and how can lighting change its value to ensure functional hybridity of the device?
7. What does the hysteretic form of the I-V curve observed for the encapsulation suggest about the accumulation of electric charge at the interphase boundaries, and how can this be relevant for the development of new electronic materials?
8. How does the introduction of SFA into the pores of the MSM-41 matrix affect the Z ᇱ in the low and high-frequency ranges, as shown in Figure 3?
9. What processes contribute to the conductivity of the MSM-41 encapsulate, and how do carriers behave in the impurity energy spectrum at different temperatures?
10. What is the magnetoresistive effect observed in the MSM-41 encapsulate, and how do the maximum values of this effect change with frequency?
11. How does the photoresistive effect in the MSM-41 encapsulate change with frequency, and what is its significance for the manufacture of ultrasensitive magnetic field sensors and lighting devices?
12. What are the values of tgδ and ε in the frequency interval of 2 X 10-3-3 X 10-1 Hz in the MSM-41 encapsulate, and what is the potential for implementing quantum storage of electrical energy?
13. How does the value of ε change under the influence of illumination, and what implications does this have for the functional hybridity of a quantum battery?
14. What is the hysteresis characteristic observed in the I-V curve of the MSM-41 encapsulate, and what causes the separation and accumulation of electric charges at the interphase boundaries?
Author Response
- What is the method used to form the heterostructured nanocomposite MCM-41, and what are the respective roles of the host matrix and the organic guest in the composite material?
Encapsulation of SFA in the pores of the MSM-41 matrix was carried out by the method of thermovacuum impregnation of the powdered material of the matrix with the obtained SFA solution. MSM-41 played the role of a dielectric matrix with a calibrated pore size and chemical resistance to various organic components. This matrix was used to ensure the nanoscale particles of the guest component and their reliable anticoagulation matrix isolation. The guest component was selected due to its extraordinary properties in macrostructure. The task was to investigate its properties in the nanoscale confined space of the dielectric matrix. The research was carried out in order to realize quantum storage of electrical energy in this type of structure.
- How is the monoporosity of the studied matrix established, and what is the maximum radius of the distribution of pores observed in the matrix?
The MSM-41 powder was purchased from Sigma-Aldrich, from whom we received detailed documentation on its parameters. In parallel, we decided to check the porosity of this matrix, for which we performed measurements by the nitrogen sorption-desorption method and calculated the pore radius distribution by the BET method. As can be seen from Figure 1, a narrow peak is observed in the vicinity of 1.42 nm.
- What are the results of the X-ray structural analysis of the matrix and encapsulate, and what does the absence of manifestation of the guest component suggest about its nature?
The X-ray structural analysis of the matrix itself, as is known because of its polycrystalline structure, shows a broad diffuse maximum. For MSM-41<SFA>, it does not show any significant changes. This may be the result of the fact that due to the nanoscale of the guest component, this technique is not able to identify them. We can see almost the same picture as for the original matrix with a slightly increased signal intensity. the nanoscale of the guest component may be a useful conclusion.
- What are the findings of the impedance studies of the encapsulate, and what does the frequency behavior of its impedance, dielectric permittivity, and tangent of the dielectric loss angle indicate about the encapsulate's properties?
Measured impedance spectra over a wide frequency range provide information about the mechanisms of electrical conductivity in a given heterogeneous medium. Thus, it was found that the electrical conductivity of this composite is greatly influenced not only by the hopping conduction of non-basic carriers through localized states near the Fermi level, but also by the processes of carrier trapping and retention in quantum wells. Quantum tunneling will also take place. Such a complex mechanism of electrical conduction will also lead to interesting polarization effects. In this case, a combination of a high dielectric constant with a dielectric loss tangent of less than 1 in the low-frequency range is obtained, which is a prerequisite for the realization of electrical energy storage through quantum effects. The greater the value of the dielectric constant in this case will be achieved, the greater the density of electrical energy can be accumulated. The emergence of quantum analogues of the currently existing electrochemical autonomous power sources is associated with this kind of materials.
- What are the potential applications of the photo- and magneto-resistive and capacitive effects observed in the encapsulate, and how can they be utilized in the preparation of highly sensitive sensors?
Due to the fact that at certain frequencies, the application of a constant magnetic field or illumination significantly changes the resistivity or dielectric constant, this encapsulant can be used as a highly sensitive sensor of these fields. Such sensors are used, for example, to read information from magnetic storage devices. Thus, the greater their sensitivity, the greater the density of recorded information. Сurrently, materials with a magnetoresistive effect are used in such sensors. This effect is associated with a change in the resistance of the sensor when a field is applied. Since the resistance is an active component, it will dissipate energy in the form of heat. In our opinion, replacing such sensors with their capacitive counterparts will avoid energy dissipation, since it will be a reactive component. Similarly, positive characteristics can be transferred to optical range sensors. The main advantage of the material we propose is its relatively low cost and ease of production compared to currently used analogues.
- How are the conditions for quantum storage of electrical energy implemented in the nanocomposite, and how can lighting change its value to ensure functional hybridity of the device?
Referring to the patent presented in this paper, a prerequisite for quantum electric energy storage is high dielectric constant with a dielectric loss tangent of less than 1 in the low-frequency range. The low-frequency range is chosen for reasons of approximation of alternating current to almost constant. In this range, it is quite easy to obtain high values of dielectric constant for ordinary materials since, classically, its behavior with frequency should decrease with increasing frequency. However, the dielectric loss tangent will be much greater than unity, making it impossible to effectively accumulate electric charge in this heterophase structure. Therefore, it is important that the value of the dielectric loss tangent is less than unity. Such conditions have been achieved in our intercalate, which leads us to conclude that this material is suitable for quantum electric energy storage.
- What does the hysteretic form of the I-V curve observed for the encapsulation suggest about the accumulation of electric charge at the interphase boundaries, and how can this be relevant for the development of new electronic materials?
It is known that electrochemical power supplies cannot be incorporated into the structure of microchips and other microelectronics elements. However, if it is possible to realize the accumulation of electrical energy, for example, at the interfaces of a heterostructure due to polarization charge conjugation, such as in a dipole, it will be possible to incorporate such sources into arbitrary microcircuits, providing them with non-volatile power. this will make a huge development in electronics. From the modeling of impedance spectra, we have shown an increase in the capacity of the interfacial barrier, and the VAR confirmed this result. The hysteresis clearly indicates the accumulation of electric charge in this structure, and the absence of maxima and minima indicates an exclusively polarization.
- How does the introduction of SFA into the pores of the MSM-41 matrix affect the Z ᇱin the low and high-frequency ranges, as shown in Figure 3?
The introduction of SFA into the pores of the MSM-41 matrix leads to an almost 5-fold decrease in in the low-frequency range (0.004 - 70 Hz) (curve 2 in Figure 3) and to its subsequent more than 4-fold increase at higher frequencies (70 - 5.5·104 Hz). The nature of the behavior of also changes, which for encapsulation MSM-41<SFA> loses monotony in the medium frequency range (0.8 - 70 Hz) and exhibits oscillatory behavior. In this case, relation (2) is no longer valid. The deviation of the experimental dependence is due to the fact that the conductivity of this encapsulate is formed not only by band carriers and jumps in localized states, but also by the processes of capture – retention – release of carriers from quantum wells. In general, the behavior of for the MSM-41<SFA> encapsulate indicates significant changes in the concentration of impurity states near the Fermi level by the guest component. In this case, shallow trap centers are also formed, which capture and hold the carriers for a time proportional to the period of the sinusoidal measurement signal, thus leading to oscillations
- What processes contribute to the conductivity of the MSM-41 encapsulate, and how do carriers behave in the impurity energy spectrum at different temperatures?
The introduction of a guest component leads to an increase in the density of states above and below the Fermi level, which leads to an increase in conductivity with increasing frequency, and also introduces deep quantum wells that trap and hold current carriers. The latter leads to the appearance of oscillations of the real part of the resistance with frequency. Pre-salting by the method of thermally stimulated discharge allows to get a more detailed view of the impurity spectrum. in accordance фs we can see, in the temperature range of 247 – 315 K, the spectrum has narrow strips with a significantly higher density of states and a well-defined miniband character. The spectrum acquires quasi-continuity at temperatures higher than room temperature starting at 315 K. Relaxation of the homocharge is observed in the entire studied temperature range.
- What is the magnetoresistive effect observed in the MSM-41 encapsulate, and how do the maximum values of this effect change with frequency?
An interesting fact is that the magnetoresistive effect is observed not only at low frequencies, but also in the high-frequency region. Its maximum values obtained at a frequency of 0.001 Hz take the value of ≈ 3 times, at a frequency of 0.8 Hz ≈ 4 times, at a frequency of 1 kHz ≈ 4 times and at a frequency of 1 MHz ≈ 25 times.
- How does the photoresistive effect in the MSM-41 encapsulate change with frequency, and what is its significance for the manufacture of ultrasensitive magnetic field sensors and lighting devices?
The photoeffect manifests itself under the condition of an increasing frequency. So, that at a frequency of 0.8 Hz it takes the value ≈ 4 times, at a frequency of 1 kHz ≈ 4 times, and at a frequency of 1 MHz ≈ 9 times.
- What are the values of tgδ and ε in the frequency interval of 2 X 10-3-3 X 10-1 Hz in the MSM-41 encapsulate, and what is the potential for implementing quantum storage of electrical energy?
The introduction of the guest component leads to an increase in ε(ω) in the frequency range of 2·10-3÷3·10-1 Hz (curves 1 and 2 in Figure 6) taking maximum values greater than 104. Such behavior of ε(ω) most likely arises due to the Maxwell-Wagner segmental polarization and additional polarization that occurs when charge carriers jump over localized states near the Fermi level.
- How does the value of ε change under the influence of illumination, and what implications does this have for the functional hybridity of a quantum battery?
Illumination also leads to an increase in ε (curve 4 in Figure 7), but the condition tgδ < 1 is fulfilled (curve 4 in Fig. 7). The photoresistive effect ρd⁄ρl in this case reaches a 3-fold value. The obtained result shows the prospects of this encapsulate MSM-41<SFA> as a material for the manufacture of a quantum battery of electric energy, and controlling its parameters with the help of light allows to create its functional hybridity.
- What is the hysteresis characteristic observed in the I-V curve of the MSM-41 encapsulate, and what causes the separation and accumulation of electric charges at the interphase boundaries?
Confirmation of the ability of encapsulate MSM-41<SFA> to accumulate an electric charge is obtained by measuring the I-V curve (Figure 8). In this case, for the MSM-41<SFA> encapsulate, the I-V curve takes a form different from the linear one, which is typical for the original desorbed matrix MSM-41, reflecting the hysteresis that is characteristic of non-Faraday energy storage devices (curve 1 in Fig. 8). This is typical for such devices as, for example, supercapacitors, which work on the effect of a double electric layer, where charge separation takes place at the solid-electrolyte boundary. In our case, the accumulation of electric charge occurs at grain boundaries due to polarization.
Thank you for the review,
і respectfully Fedir Ivashchyshyn
Reviewer 3 Report
The article investigates the electrical properties of a nanocomposite material obtained by impregnation with fulvic acid (SFA) of nanoporous silicon oxide (SiO2) of the MCM-41 brand.
The big disadvantage of the article is the lack of studies of the obtained material by electron microscopy (SEM or TEM). These methods would make it possible to clarify the structure of the nanocomposite material, since the question arises how deeply SFA has penetrated into the pores of SiO2. The pore size is in the range of no more than 2 nm. The viscosity of SFA is unknown. In my opinion, the SFA only clogged the pores, but did not penetrate them. These studies must be carried out. Then the article will improve significantly.
Remarks:
L40-65. The introduction needs to be rewritten, since it does not take into account modern technological trends in the formation of nanostructured materials for energy storage devices formed according to the "host-guest" principle.
L87-94. It is necessary to describe the manufacturing technology of SFA and specify the brand and manufacturer of the materials used.
L99. Which substance powder (SFA or SiO2) was heated to 140 oC?
L108, 112,115. It is necessary to specify the country and the manufacturer of the devices.
L112. It is necessary to specify the type and country of the manufacturer of the diffractometer that was used.
L129. It is necessary to specify the type and country of the manufacturer of the device to study the method of thermally stimulated discharge.
L139. Misspell.
L168,179,180,189 Error in formula numbering.
L199. It is necessary to specify the frequency range.
L208-210. It is necessary to write an expression for the magnetoresistive effect.
L217-219. It is necessary to write an expression for calculating the photo effect.
L292-318. The authors should evaluate the accumulated electrical energy. Then it is possible to compare the parameters of energy storage devices based on MSM-41 (SFA) with other known energy storage devices.
Author Response
The big disadvantage of the article is the lack of studies of the obtained material by electron microscopy (SEM or TEM). These methods would make it possible to clarify the structure of the nanocomposite material, since the question arises how deeply SFA has penetrated into the pores of SiO2. The pore size is in the range of no more than 2 nm. The viscosity of SFA is unknown. In my opinion, the SFA only clogged the pores, but did not penetrate them. These studies must be carried out. Then the article will improve significantly.
Thank you for the valid remark. I agree that for a more detailed analysis of this encapsulate, additional research should be conducted. However, I know for sure that it will not be possible to do this with the help of SEM electron microscopy. We conducted such studies and due to the weak conductivity of this material, the magnification and resolution cannot be achieved. Examination by electron microscopy TEM has certain caveats for the preparation of this type of materials for examination. We have not yet conducted such studies, but we plan to do so in the future.
L40-65. The introduction needs to be rewritten, since it does not take into account modern technological trends in the formation of nanostructured materials for energy storage devices formed according to the "host-guest" principle.
The introduction has been improved. Replaced references 10, 11 and 12 with newer ones, and added references 13 and 14 on sensing. Removed self-citations, leaving only reference 17, which was also replaced by a newer one.
L87-94. It is necessary to describe the manufacturing technology of SFA and specify the brand and manufacturer of the materials used.
The production technology of SFA is the author's part of the author's collective and it is patented. Сlarification of the technology and brand of precursor was added to the text.
L99. Which substance powder (SFA or SiO2) was heated to 140 °C?
Тhe powder MSM-41 (SiO2) was heated to a temperature of 140 °C. Clarification added in the text.
L108, 112,115. It is necessary to specify the country and the manufacturer of the devices.
Added to the text – USA
L112. It is necessary to specify the type and country of the manufacturer of the diffractometer that was used.
Added to the text – The measurement was carried out using a DRONE-3 (USSR).
L129. It is necessary to specify the type and country of the manufacturer of the device to study the method of thermally stimulated discharge.
Added to the text – The thermally stimulated discharge spectra were recorded in the mode of short-circuited contacts. To measure the current, the B7-30 (USSR) was used, which is characterized by an input impedance of 1015 Ohm.
L139. Misspell.
Сorrected
L168,179,180,189 Error in formula numbering.
Сorrected
L199. It is necessary to specify the frequency range.
Added to the text – in the entire studied range.
L208-210. It is necessary to write an expression for the magnetoresistive effect.
Added to the text – The magnetoresistive effect was calculated by the formula . – value of resistance Z' under normal conditions, and – value of resistance Z' when a constant magnetic field is applied.
L217-219. It is necessary to write an expression for calculating the photo effect.
Added to the text – The photoeffect was calculated by the formula . – is the resistance value Z' under normal conditions in the dark, and – is the resistance value Z' under illumination.
The photocapacitance effect in this case reaches a 3-fold value. – is the value of the dielectric constant under normal conditions in the dark, and – is the value of the dielectric constant under illumination.
L292-318. The authors should evaluate the accumulated electrical energy. Then it is possible to compare the parameters of energy storage devices based on MSM-41 (SFA) with other known energy storage devices.
Added to the text – The essence of the concept is to store electrical energy by polarizing electrons rather than in the form of electrolyte ions. Known dielectrics in which this possibility was opened ε ≤ 100. The author of the patent [16] theoretically showed that in the case of increasing ε to ~106 it is possible to achieve a higher energy density 1 MJ/kg. At the moment, these are theoretical calculations that allow us to see the prospects for the development of this area of research.
Thank you for the review,
і respectfully Fedir Ivashchyshyn
Round 2
Reviewer 2 Report
Accept after corrections to text editing
Author Response
The text of the article was revised and the English language was improved. The relevant changes are marked in blue.
Reviewer 3 Report
The authors responded to a number of comments. However, some important questions were not answered.
The main drawback of the manuscript has not been eliminated. There are no studies of the obtained material by electron microscopy (SEM or TEM) methods in the manuscript. In my opinion, the SFA only clogged the pores, but did not penetrate them. Atomic force microscopy (AFM) can also help here. When using very sharp cantilevers, you can get the necessary permission. I would like the authors of the manuscript to have the desire and opportunity to do this. Then the manuscript will improve significantly.
Unfortunately, the introduction was improved with only one phrase about "...quantum amplification of sensory sensitivity to external physical fields.". However, the prospects of using the accumulation of electrical energy due to quantum-dimensional effects have not yet been sufficiently shown.
There are a number of important and not fully explained issues:
The authors did not explain the mechanism of energy storage at the boundaries of the SFA and MSM-41 section in any way.
How does the composite nanomaterial proposed in the manuscript differ from nanomaterials used for energy storage in supercapacitors? Only by the fact that the effect of thermally stimulated discharges was detected at negative temperatures? Maybe these are current surges in the pores in which water vapor is present. For clarification, I would like to see studies of thermally stimulated discharge currents for MSM-41 in Fig.4.
I would like the authors to evaluate the stored energy based on the data presented in Figure 8 and compare it with the characteristics of known energy storage devices. But the authors did not do this.
Remarks
L94-100. It is necessary to describe the manufacturing technology of SFA. The authors did not do this.
L329. What do the authors mean by the expression:
"The essence of the concept is to store electrical energy by polarizing electrons rather than in the form of electrolyte ions"?
L375-376. The output is incorrect. The magnetic field used by the authors is very large (500 times larger than the earth's magnetic field), and the response to its impact is only 25. The lighting used corresponds to the maximum solar radiation, and the response to its impact is only 9.
Authors should carefully proofread the text. There are typos in the text.
L286, Table.1. The designation of the studied material is written with an error.
L319, 321. An error in the drawing number.
Author Response
Reviewer 3.
Answers to the reviewer's comments.
The authors responded to a number of comments. However, some important questions were not answered.
The main drawback of the manuscript has not been eliminated. There are no studies of the obtained material by electron microscopy (SEM or TEM) methods in the manuscript. In my opinion, the SFA only clogged the pores, but did not penetrate them. Atomic force microscopy (AFM) can also help here. When using very sharp cantilevers, you can get the necessary permission. I would like the authors of the manuscript to have the desire and opportunity to do this. Then the manuscript will improve significantly.
Atomic force microscopy is well suited to the study of thin films, namely their surface structure and relief. However, this method is difficult to apply to powdered material. Preparing a test sample would require quite a bit of work and solving a number of problems. This, in turn, requires time and the ability to conduct such a study. We completely agree, but I am sorry to say that, despite our desire, we are currently unable to conduct such research. The authors and I realize their importance, but we cannot do it now.
Unfortunately, the introduction was improved with only one phrase about "...quantum amplification of sensory sensitivity to external physical fields.". However, the prospects of using the accumulation of electrical energy due to quantum-dimensional effects have not yet been sufficiently shown.
Added to the text – A more modern work in this area is the theoretical model of a supercapacitor presented by scientists from Cornell University [18]. The researchers predicated two chains in their model: one chain deals with electrons whereas the other chain primarily deals with holes. Тhis study confirmed that their two-chain model demonstrated a significantly enhanced level of quantum capacitance.
and – [18] Ferraro, D.; Andolina, G. M.; Campisi, M.; Pellegrini, V.; & Polini, M. Quantum supercapacitors. Physical Review B 2019, 100(7). doi:10.1103/physrevb.100.075433
There are a number of important and not fully explained issues:
The authors did not explain the mechanism of energy storage at the boundaries of the SFA and MSM-41 section in any way.
The explanation is given in the text, it may be difficult to see:
In our case, the accumulation of electric charge occurs at grain boundaries due to polarization.
And the polarization itself is explained earlier in the text:
As it can be seen, the introduction of the guest component leads to an increase in ε(ω) in the frequency range of 2·10-3÷3·10-1 Hz (curves 1 and 2 in Figure 6) taking maximum values greater than 104. Such behavior of ε(ω) most likely arises due to the Maxwell-Wagner segmental polarization and additional polarization that occurs when charge carriers jump over localized states near the Fermi level. Maxwell-Wagner polarization occurs in the presence of macrodipoles that arise in the vicinity of charged defects, as well as at heteroboundaries of the structure. In this case, instead of one group of particles capable of relaxation polarization with one defined relaxation time τ, there is a large number of relaxers with different values of the time τ. This is confirmed by the dependence of tgδ(ω) (curves 1 and 2 in Figure 7). For the original matrix MSM-41 (curve 1 in Figure 7), a pronounced main peak is observed, which corresponds to the main relaxation time for this material. On the other hand, in the case of MSM-41<SFA> encapsulation, a tgδ(ω) oscillation is obtained, which clearly indicates the existence of a wide range of relaxation times caused by relaxation centers introduced by the guest component.
How does the composite nanomaterial proposed in the manuscript differ from nanomaterials used for energy storage in supercapacitors? Only by the fact that the effect of thermally stimulated discharges was detected at negative temperatures? Maybe these are current surges in the pores in which water vapor is present. For clarification, I would like to see studies of thermally stimulated discharge currents for MSM-41 in Fig.4.
Added to the text – No thermal stimulated discharge currents were recorded for the original MSM-41.
The main challenge now lies in low quantum capacitance, which is a direct result of the shortage of quantum states near the Fermi level. As a critical component in the determination of the electrical and thermal properties of solids, the Fermi level changes as a solid is warmed and more relevantly, when electrons are added or taken from the solid. That is, the task is to raise the dielectric constant and ensure a high density of states near the Fermi level. To a greater extent, a guest component was introduced into the dielectric matrix to achieve the latter. To realize quantum-dimensional effects, the size of the guest component should be nanoscale, which was to be provided by a monoporous matrix.
As for carbon materials for supercapacitors, their dielectric constant does not exceed 100 and charge separation occurs in the form of electrons in the solid and positive ions in the electrolyte. Ions are rather large particles in relation to electrons, so this storage device has certain limitations in its miniaturization and power consumption.
To remove water from the sample, it was thoroughly dried until it stopped losing weight.
I would like the authors to evaluate the stored energy based on the data presented in Figure 8 and compare it with the characteristics of known energy storage devices. But the authors did not do this.
Added to the text – In order to correctly compare our results with the data presented in [17], we calculated the specific capacitance for both cases. The paper indicates such parameters as the specific energy W = 1.66 MJ/kg when the voltage on the capacitor covers reaches 1.37·105 V. The energy of the capacitor is determined by Eq:
(6)
where C is the capacitance of the capacitor, U is the voltage across the capacitor.
The capacitance will be determined accordingly:
(7)
As a result of simple calculations, the specific capacitance C = 1.77·10-4 (F/kg) = 1.77·10-7 (F/g) = 0.177 (μF/g).
Accordingly, we can calculate the specific capacity of our material using the formula:
(8)
where q is the accumulated charge on the capacitor linings, U is the voltage across the capacitor, and m is the mass of the sample under test.
From the presented I-V current in Figure 8, we can determine the value of q by integrating the current over time
(9)
for measurements under normal conditions, in a constant magnetic field, and under illumination. Substituting these values into formula (8), we obtain the following values of the specific capacitance of our material Сnc = 0,053 μF/g, Cm = 1,260 μF/g, Cl = 0,56 μF/g. So, from the data presented, we can see that our material has more than twice the specific capacitance theoretically proposed in [17], but it has functional hybridity and takes on a higher value when illuminated, and a much higher value when a constant magnetic field is applied.
Remarks
L94-100. It is necessary to describe the manufacturing technology of SFA. The authors did not do this.
Added to the text – The peculiarity of this SFA is that it has a chemical composition close to the natural one, i.e. containing N and S atoms, by introducing thiourea (Sigma Aldrich) into the reaction mixture, which is a source of nitrogen and sulfur in the synthesized fulvic acid. The choice of pyrocatechin and thiourea as precursors was due to their ability to enter into a substitution reaction, the product of which simultaneously contains a phenolic nucleus and a thiocarbamide residue.
To prepare the reaction mixture, a 2.2 g weight of pyrocatechin in a polyethylene bag was placed in a 1 L Kjeldahl flask, and then a solution containing 3.2 g of NaOH, 0.76 g of thiourea, and distilled water was carefully added to the wall to a total volume of 100 mL. The air was purged from the flask by slowly blowing an excess of pure electrolytic oxygen.After filling the reactor flask with oxygen, it was sealed and mechanical shaking was started, which led to the mixing of the reagents and the start of the reaction. The reaction was carried out until the oxygen consumption slowed down to less than 0.01 mol O2 per 1 mol pyrocatechin per hour.
After completion of the oxidation reaction, the reaction mixture was passed through an ion-exchange column filled with KU-2-8 cationite in the Η-form. The eluate was started to be removed when a brown color appeared and was completed when the color changed to light red. The volume of the obtained fraction was equal to the volume of the reaction mixture. The liquid was heated on an electric stove to remove excess dissolved carbon dioxide. The resulting solution had a pH of 2.8.
L329. What do the authors mean by the expression:
"The essence of the concept is to store electrical energy by polarizing electrons rather than in the form of electrolyte ions"?
Replaced by –The essence of the concept is to store electrical energy using polarized electrons and holes, rather than in the form of electrons and electrolyte ions as in an electrochemical supercapacitor.
L375-376. The output is incorrect. The magnetic field used by the authors is very large (500 times larger than the earth's magnetic field), and the response to its impact is only 25. The lighting used corresponds to the maximum solar radiation, and the response to its impact is only 9.
Usually in the literature, the formula for the magnetoresistive effect is given in the following form:
- 100%
i.e., usually the value does not exceed 100%. In our case, we got a high value of this effect as much as 25 times.
Аor example, in the article linked to [14]:
Akin, M.; Pratt, A.; Blackburn, J.; & Dietzel, A. Paper-Based Magneto-Resistive Sensor: Modeling, Fabrication, Characterization, and Application. Sensors 2018, 18(12), 4392. doi:10.3390/s18124392
The maximum value of the magnetoresistive effect of 0.4 % was obtained at a magnetic field intensity of 2.7 kA/m. The field is 100 times larger, but the effect itself is more than 1000 times larger.
I also added a link [15] to the article:
Stepanov, G. V.; Bakhtiiarov, A. V.; Lobanov, D. A.; Borin, D. Yu.; Semerenko, D. A.; Storozhenko, P. A. Magnetoresistive and magnetocapacitive effects in magnetic elastomers. SN Applied Sciences 2022, 4, 178. doi: 10.1007/s42452-022-05068- y
At 330 mT, a sample of magnetic elastomer shows a magnetoresistive effect of 3200 % (32 times). 330 mT = 330 GA/m. (In our case, at 220 kАm, the magnetoresistive effect is 25 times)
At 330 mT, a sample of magnetic elastomer shows a capacitance higher by more than 30 times. (In our case, at 220 kАm, the photocapacitance effect is 5 times)
The permanent magnetic field (220 kАm) strength of the permanent magnet we use is not high.
Authors should carefully proofread the text. There are typos in the text.
Thank you. We have reviewed the text for typos.
L286, Table.1. The designation of the studied material is written with an error.
Thank you. Сorrected
L319, 321. An error in the drawing number.
Thank you. Сorrected
Thank you for the review,
і respectfully Fedir Ivashchyshyn
